# Berberine Rescues D-Ribose-Induced Alzheimer‘s Pathology via Promoting Mitophagy

**DOI:** 10.3390/ijms24065896

**Published:** 2023-03-20

**Authors:** Chuanling Wang, Qian Zou, Yinshuang Pu, Zhiyou Cai, Yong Tang

**Affiliations:** 1Department of Histology and Embryology, School of Basic Medicine, Chongqing Medical University, No. 1 Yixueyuan Road, Yuzhong District, Chongqing 400016, China; 191048@cqmu.edu.cn; 2Chongqing Key Laboratory of Neurodegenerative Diseases, No. 118 Xingguang Avenue, Liangjiang New Area, Chongqing 401147, China; 3Department of Neurology, Chongqing General Hospital, No. 118 Xingguang Avenue, Liangjiang New Area, Chongqing 401147, China

**Keywords:** Alzheimer’s disease, BBR, D-ribose, mitophagy, cognition

## Abstract

Mitochondrial dysfunction is considered an early event of Alzheimer disease (AD). D-ribose is a natural monosaccharide that exists in cells, especially in mitochondria, and can lead to cognitive dysfunction. However, the reason for this is unclear. Berberine (BBR) is an isoquinoline alkaloid that can target mitochondria and has great prospect in the treatment of AD. The methylation of PINK1 reinforces the burden of Alzheimer’s pathology. This study explores the role of BBR and D-ribose in the mitophagy and cognitive function of AD related to DNA methylation. *APP/PS1* mice and *N2a* cells were treated with D-ribose, BBR, and mitophagy inhibitor Mdivi-1 to observe their effects on mitochondrial morphology, mitophagy, neuron histology, AD pathology, animal behavior, and PINK1 methylation. The results showed that D-ribose induced mitochondrial dysfunction, mitophagy damage, and cognitive impairment. However, BBR inhibition of PINK1 promoter methylation can reverse the above effects caused by D-ribose, improve mitochondrial function, and restore mitophagy through the PINK1–Parkin pathway, thus reducing cognitive deficits and the burden of AD pathology. This experiment puts a new light on the mechanism of action of D-ribose in cognitive impairment and reveals new insights in the use of BBR for AD treatment.

## 1. Introduction

Alzheimer’s disease (AD) is a chronic progressive neurodegenerative disease classically characterized by memory loss, cognition impairment, and progressive β-amyloid (Aβ) and phosphorylated tau accumulation, ultimately causing loss of neurons and synapses, brain atrophy, and even death [1]. Between 2000 and 2019, the number of deaths due to AD increased by more than 145% [1], and AD has become an important global public health problem. Despite extensive basic and clinical research, no effective therapeutic strategies for AD have been found. Therefore, it is an urgent task to explore the pathogenesis and find novel biomarkers and therapies of AD. 

D-ribose is found in all living cells and obtained from diet and endogenous synthesis. It is a key component of ribonucleic acid (RNA), acetyl coenzyme A, and adenosine triphosphate (ATP) [2]. It has been reported that the level of D-ribose in AD patients was significantly higher than that of cognitively intact individuals of similar age, which might be a new potential diagnostic biomarker for AD [3]. D-ribose is danger to health if consumed in excess [4]. For example, chronic overconsumption of D-ribose resulted in depression/anxiety and spatial memory impairment in mice [2]. High levels of D-ribose in serum and urine are involved in diabetic encephalopathy [5]. The administration of D-ribose produced high levels of advanced glycation end products (AGEs), including the hyperphosphorylation of Tau protein, in the brain of *C57BL/6* mice and *N2a* cells [6], suggesting that high levels of D-ribose intake may have a damaging effect on the nervous system. However, the specific mechanism needs further study.

Cells produce D-ribose, which is essential for ATP production in mitochondria [7]. Mitochondria produce 99% of ATP, which is the main energy source of a highly active brain, and so mitochondrial quality control is crucial. The PINK1 (PTEN induced kinase 1)–Parkin (parkin RBR E3 ubiquitin protein ligase) pathway is a classic pathway of mitophagy, which is the key to maintaining mitochondrial homeostasis, and mitochondrial dysfunction is considered to be an early event of AD. Studies have shown that mitophagy in AD patients and AD models is significantly reduced [8,9]; Aβ and P-Tau hinder the PINK1–Parkin pathway, increase the production of ROS, destroy the mitochondrial membrane, and aggravate the mitochondrial structural and functional abnormalities [10,11], suggesting that the neuropathology of AD forms a vicious cycle with mitophagy disorder. Interestingly, D-ribose was considered to inhibit the autophagy of AD and induce the generation of Aβ [12], although whether it affects mitophagy remains to be studied.

Berberine (BBR) has a neuroprotective effect that can cross the blood–brain barrier and inhibit the formation of amyloid plaque deposits and neurofibrillary tangles, significantly improving memory and cognitive dysfunction [13]. Studies have shown that BBR activates PINK1–Parkin-dependent mitophagy, inhibits mitochondrial damage, reduces ROS generation, protects cardiac function, and reduces kidney injury [14,15]. Additionally, BBR can alleviate mitochondrial abnormalities in neurons by maintaining mitochondrial membrane potential, increasing mitochondrial density and length, and improving mitochondrial movement and transport [16], indicating that BBR has a special protective effect on mitochondria. Our previous work has preliminarily proved that BBR can enhance autophagy and reduce D-ribose-induced Aβ [12]; however, whether this is achieved by protecting mitochondrial function remains unknown.

DNA methylation is the most common epigenetic modification studied in AD, without changing the DNA sequence but regulating gene expression, which is particularly sensitive to environmental stimuli and affects cognitive function [16]. Studies have revealed that PINK1 methylation reinforces the burden of Alzheimer’s pathology [17]. Interestingly, BBR has been shown to participate in the regulation of gene methylation [18]. Therefore, this study explores the effects of D-ribose on the mitophagy and pathological and cognitive changes in AD models, and observes whether BBR resists the effects of D-ribose by affecting PINK1 methylation, providing a new experimental basis for D-ribose-induced cognitive dysfunction and BBR for the treatment of AD.

## 2. Results

### 2.1. Berberine Ameliorates D-Ribose-Induced Mitochondrial Dysfunction via Promoting Mitophagy

Mitochondrial membrane potential (ΔΨm), ROS production, and Cytc release can reflect the functional status of mitochondria and are closely linked to apoptosis. Flow cytometry analysis of ΔΨm showed that the D group had weakened red fluorescence, enhanced green fluorescence, and decreased the ratio of the JC-1 aggregate/JC-1 monomer fluorescence compared with the control group, showing that D-ribose caused a significant decrease in ΔΨm (Figure 1A–C). However, ΔΨm increased after the treatment with BBR (DB group), but decreased again after the addition of Midivi-1 (DBM group) or siRNA knockdown of PINK1 (DBS group). Similarly, the results of ROS production (Figure 1D–F), Cytc expression (Figure 1G,H) and apoptosis (Figure 1I,J) in each group showed the same trend as the results of ΔΨm, suggesting that D-ribose-induced a decrease in ΔΨm, overload of ROS, Cytc release, and ultimately lead to cell death. Treatment with berberine can reverse D-ribose-induced mitochondrial dysfunction via promoting mitophagy.

### 2.2. Berberine Revives D-Ribose-Induced Mitophagy Dysfunction through PINK1–Parkin Pathway

Mitophagy is the key to maintaining mitochondrial quality, and PINK1–Parkin-induced mitophagy can revive dysfunctional or damaged mitochondria. Therefore, mitophagy was investigated to further clarify the protective role of berberine in Alzheimer’s disease. Electron microscopy showed that autophagy vesicles were observed in *N2a* cells of the Ctrl and DB groups. Moreover, mitochondria of the Ctrl and DB groups were basically normal with uniform matrix in the hippocampus of *APP/PS1* mice, while mitochondria of the D and DBM groups were significantly swollen and cristae were ruptured (Figure 2A). These results suggest that berberine can protect mitochondrial structure damaged by D-ribose through mitophagy. 

Mito Tracker (green) and lyso Tracker (red) were then used to observe the fusion of mitochondria and autolysosomes. The co-localization results showed that berberine increased the co-localization of mitochondria and lysosomes through the mitophagy, which counteracted the effect of D-ribose (Figure 2B,C). In addition, compared with the Ctrl group, the co-localization of LC3A/B and TOMM20 (Figure 2F,G), PINK1 and Parkin (Figure 2D,E) in the D group was significantly reduced, which blocked PINK1–Parkin-induced mitophagy. Berberine reversed the above results caused by D-ribose, suggesting that berberine revives D-ribose-induced mitophagy dysfunction through the PINK1–Parkin pathway.

The detection of the PINK1–Parkin-mitophagy pathway showed that D-ribose could inhibit the expression of the key mitophagy-associated proteins PINK1, Parkin, and LC3B, both in vitro (Figure 3A–F) and in vivo (Figure 3G–J). Furthermore, P62 was cleared and the expression of LC3B increased after berberine intervention, suggesting that the PINK1–Parkin-mitophagy pathway was activated by berberine.

### 2.3. Berberine Mitigates Alzheimer’s-Like Pathology Induced by D-Ribose via Promoting Mitophagy

Compelling studies have shown that impaired mitochondrial function occurs earlier than pathological changes, accelerating the deposition of Aβ and hyperphosphorylation of Tau protein in AD models and patients [10,19,20]. This study showed that the expression of Aβ, P-TAU (T205), P-TAU (T231) and P-TAU (S396) in the hippocampus and cortex of *APP/PS1* mice in the D group was significantly increased compared with the control group (Figure 4A). Compared with the D group, the above indexes in DB group were decreased. After addition of mitophagy inhibitor Midivi-1 in DBM group, the expression of these proteins was significantly increased again. Meanwhile, the results of immunohistochemical staining also showed the same effect (Figure 4B), suggesting that D-ribose can increase the burden of AD pathology, while berberine reduced the pathological changes of AD by promoting mitophagy.

Neuron damage is considered to be one of the important characteristics of AD, so we performed H & E staining and Nissl staining to investigate the morphological alteration and neuron loss, respectively. As shown in Figure 5A, the neurons in the control group were arranged neatly, with full and normal morphology, clear membrane boundary, pale red cytoplasm, blue nucleus and clear nucleolus. Compared with the control group, the neurons in DB group were arranged irregularly, with loss of morphology, enlarged gap, nuclear pyknosis, and cellular vacuolization accompanied by necrosis. However, BBR can rescue the effect of D-ribose. Compared with the D group, the morphology of neurons in DB group was greatly improved; However, compared with the DB group, neurons in the DBM group with midivi-1 showed nuclear pyknosis and cellular vacuolization.

The same trend as H & E staining was observed in Nissl staining (Figure 5B). In the control group, neurons were complete in structure and rich in blue Nissl bodies, while the D group was incomplete in structure, vague in shape, and uneven in Nissl staining in cytoplasm; Compared with the D group, the morphology of neurons in DB group with BBR was plump and regular, and Nissl bodies were abundant. However, compared with DB group, Nissl staining in DBM group was not uniform. These results suggest that BBR can improve the cerebral cortex and hippocampal pathological injuries cause by D-ribose through mitophagy.

### 2.4. BBR Ameliorates Cognitive Impairment of APP/PS1 Mice Induced by D-Ribose via Mitophagy

The literature shows that the level of D-ribose in serum and urine is negatively correlated with cognitive function and induce anxiety-like behavior [2,21,22]. Therefore, the cognition and mood were checked through behavioral assessment in *APP/PS1* mice.

The Morris water maze test was used to study the spatial learning and memory. Compared with the control group, the mice in the D group had longer escape latency, shorter target quadrant time, and were more likely to deviate from the platform position; After adding BBR, the escape latency in DB group was significantly shorter than that in the D group, meanwhile, the time in the platform and target quadrant was longer; However, after adding the mitophagy inhibitor Midivi-1, the DBM group showed similar results with the D group, and the average speed of each group was not different (Figure 6A–F). 

The two-way shuttle (TWS) avoidance task can be used to evaluate the memory of mice [23]. The trend was similar to that of MWM. Compared with the control group, the shuttle times in the D group were significantly reduced and the escape latency was increased. The addition of BBR could significantly reverse the effect of D-ribose, but the DBM group showed fewer shuttle times and longer escape latency after the addition of Midivi-1 (Figure 6G,H). The above results suggest that D-ribose destroys the spatial cognitive, learning, and memory abilities of mice, and BBR can antagonize the D-ribose effect through mitophagy.

Anxiety was assessed using a field test. Compared with the control group, the mice of the D group spent a longer time in the periphery and a shorter time and distance in center, showing more anxiety-like behavior. The effect of D-ribose was not affected after taking BBR or Midivi-1 (Figure 6I–O), suggesting that BBR could not relieve the anxiety induced by D-ribose in *APP/PS1* mice.

### 2.5. BBR Inhibits PINK1 Promoter Methylation

The above results suggest that BBR can protect mitochondrial function through the PINK1–Parkin pathway against a D-ribose inducer, but the specific regulation of mitophagy remains unclear. In general, DNA hypermethylation in promoter regions can inhibit gene expression. Literature has shown that the CpGs methylation of PINK1 is closely related to the burden of AD pathology [17], and both glycoylation and BBR are involved in epigenetic regulation [18,24]. Therefore, the level of PINK1 promoter methylation was detected in this study. The results showed that the methylation of PINK1 promoter in the D group was increased compared with the control group after adding D-ribose. However, BBR (DB group) could significantly reduce methylation of PINK1 promoter, which has the same effect as DNA methylation inhibitor 5-Azacytidine (DA group) (Figure 7A,B). Meanwhile, in both cell and animal models, compared with the control group, the D group had increased expression of DNMT1, but decreased expression of PINK1. Compared with the D group, PINK1 was overexpressed, while DNMT1 was reduced in the DB group and DA group (Figure 7C–G). Increased promoter methylation and reduced expression of PINK1 was observed, suggesting that the promotion of PINK1 promoter methylation by D-ribose was one of the reasons for the decreased expression of PINK1, while BBR could reverse the effect of D-ribose and lead to increased PINK1 expression, which promoted mitophagy and maintained mitochondrial homeostasis.

## 3. Discussion

Progress in the understanding of the extent and role of berberine in Alzheimer’s disease has increased substantially in the past decade. Numerous variables have been discovered in which berberine is involved in the Alzheimer’s pathophysiological processes, such as senile plaques, neurofibrillary tangles, acetylcholinesterase enzyme, oxidative stress, neuroinflammation, and others [13], leaving us with a very interesting question in its pharmacological mechanism and neuroprotective role. This study showed that berberine can reverse D-ribose-induced mitochondrial dysfunction while berberine revives D-ribose-induced mitophagy dysfunction through the PINK1–Parkin pathway. Berberine mitigates cognitive impairment and Alzheimer’s-like pathology induced by D-ribose via promoting mitophagy, through which berberine inhibits PINK1 promoter methylation and promotes its expression. This study may provide potential interventions centered on the regulation of PINK1–Parkin-dependent mitophagy and offer therapeutic strategies for the treatment of Alzheimer’s disease.

Berberine, an isoquinoline alkaloid, has multiple pharmacological effects, including its purported antioxidant and antimicrobial properties for a series of diseases, such as obesity [25], diabetes [26], hyperlipidemia [27], heart failure [14], *H. pylori* infection [28], and colonic adenoma prevention [29]. Numerous hypotheses about the ways in which berberine may help with Alzheimer’s disease also have been raised, including retarding oxidative stress and neuroinflammation [13,30], limiting the pathogenesis of extracellular amyloid plaques and intracellular neurofibrillary tangles [13,31]. Previously, our group has reported that BBR attenuates cognitive deficits and limits hyperphosphorylation of tau via inhibiting the activation of NF-kappaB signaling and retarding oxidative stress and neuro-inflammation [32]. Berberine alleviates amyloid-beta pathology in the brain of *APP/PS1* mice via inhibiting beta/gamma-secretases activity, enhancing alpha-secretases [33], and activating LKB1/AMPK signaling [34]. BBR ameliorates D-ribose-induced amyloid-beta pathology via inhibiting mTOR/p70S6K signaling and improves spatial learning and memory [12]. However, more detailed investigations are warranted to clarify the role of berberine in limiting Alzheimer’s-like pathologies.

Advanced glycation end products (AGEs) is involved in the onset and exacerbation of Alzheimer’s disease while numerous studies favored that glycation help extracellular β-amyloid deposition as neuritic plaques and intracellular accumulation of hyperphosphorylated tau as neurofibrillary tangles [35,36]. The interactions of AGEs with the receptors of AGEs (RAGE) result in β-amyloid deposition and accumulation of hyperphosphorylated tau and further downstream inflammatory cascade events in Alzheimer’s pathogenesis [36,37]. A growing body of research shows that D-ribose-induced ribosylation and excessive AGE production play an important role in the formation of amyloid plaques and neurofibrillary tangles [3,12]. Recently, we have found that BBR mitigates D-ribose-induced amyloid pathology while BBR promotes autophagic lysosomal pathway by suppressing mTOR/p70S6K signaling, regulates the activity of autophagy-related proteins Beclin1, Atg3, and LC3B, and promotes Aβ clearance [12]. However, the underlying mechanism by which BBR inhibits D-ribose-induced amyloid pathology involving ribosylation-induced autophagy dysfunction remains unclear.

An abundant literature links the modulation of autophagy to altered Alzheimer’s pathogenesis [38,39]. Mitophagy, a specialized form of macroautophagy, selectively degrades damaged and dysfunctional mitochondria which contribute to normal aging and a wide spectrum of age-related diseases [40], including Parkinson’s disease and Alzheimer’s disease [41,42]. Hence, maintaining a healthy mitophagy status in aged individuals might be a beneficial strategy. This study has discovered that berberine ameliorates D-ribose-induced mitochondrial dysfunction through enhancing mitophagy while D-ribose-induced ΔΨm increase, ROS decreases, Cytc release less and ultimately cell death occurs after the treatment with berberine.

Mitophagy has been physiologically responsible for mitochondrial quality control and mitochondrial ROS balance by regulating mitochondrial trafficking and mitochondrial quality control and removing damaged mitochondria [43,44]. Growing evidence supports the contribution of mitophagy impairment to Alzheimer’s disease [45]. Defective mitophagy is thought to be responsible for the accumulation of damaged mitochondria, which leads to cellular dysfunction and death in Alzheimer’s disease [46]. PTEN-induced putative kinase 1 (PINK1) and Parkin are involved in a common pathway to regulate mitophagy and mitochondrial dynamics. PINK1–Parkin mainly regulates ubiquitin-dependent mitophagy to ensure the maintenance of mitochondrial health and homeostasis, and to deliver defective mitochondria to the lysosome for degradation [44,47]. The current knowledge has displayed the molecular mechanisms underlying mitophagy dysregulation in Alzheimer’s disease, especially in relation to the PINK1–Parkin-mediated mitophagy [46,48]. We found that berberine alleviates Alzheimer’s-like pathology induced by D-ribose via promoting mitophagy while berberine revives the PINK1–Parkin pathway. However, how berberine influences the PINK1–Parkin pathway is not well understood.

There is growing evidence for the prominent role of DNA methylation (DNAm) in Alzheimer’s disease while DNA hypermethylation in promoter regions can inhibit gene expression [49,50]. Studies have demonstrated that the CpGs methylation of PINK1 reinforces the burden of Alzheimer’s pathology [17,51]. This study found that BBR reversed the level of PINK1 promoter methylation by D-ribose and enhanced PINK1 expression. Therefore, BBR reversed the level of PINK1 promoter methylation by D-ribose and enhanced PINK1 expression. BBR ameliorates Alzheimer’s pathology and cognitive impairment of *APP/PS1* mice induced by D-ribose through inhibiting PINK1 promoter methylation.

This study found that berberine ameliorates D-ribose-induced mitochondrial dysfunction via promoting mitophagy, and second, berberine promotes D-ribose-induced mitophagy dysfunction through the PINK1–Parkin pathway. Third, berberine alleviates the burden of Alzheimer’s pathology induced by D-ribose via promoting mitophagy and improves cognitive impairment. Finally, this study discovered that berberine rescues D-ribose-induced Alzheimer’s pathology via inhibiting PINK1-promoter methylation (Figure 8). As BBR has been used clinically for many years, it might have applicable potential in regulating mitophagy and improving Alzheimer’s pathology and cognitive impairment.

## 4. Materials and Methods

### 4.1. Animals, Cell Lines, and Reagents

*APP/PS1* mice were purchased from Cavens Biogle (Suzhou, China) Model Animal Research Co., Ltd. and were raised in the Department of Laboratory animal center of Chongqing Medical University. The study was conducted in accordance with the Declaration of Helsinki, and approved by the Institutional Ethics Committee of Chongqing Medical University (date of approval, 26 February 2022).

The murine neuroblastoma cell line *Neuro2a* (*N2a*) was purchased from Procell Life Science and Technology Co., Ltd. (CL-0168, Wuhan, China), and was maintained in Dulbecco’s Modified Eagle Medium: Nutrient Mixture F-12 (DMEM/F12, Gibco, Carlsbad, CA, USA) with 10% fetal bovine serum (FBS, Invigentech, Irvine, CA, USA) in a 5% CO_2_ incubator at 37 °C.

The reagents and antibodies used in this study were as follows: BBR (Chengdu Jinghua Pharmaceutical Co., Ltd., Sichuan, China), D-ribose (V900389, Sigma-Aldrich, St. Louis, MO, USA), Mdivi-1 (HY-15886, MedChemExpress, Monmouth Junction, NJ, USA), 5-Azacytidine (HY-10586, MedChemExpress, Monmouth Junction, NJ, USA). CST and Abcam antibodies come from MA, USA. Proteintech and SAB antibodies come from Chicago and Maryland, respectively. Beyotime and Bioss antibodies came from Shanghai and Beijing, China, respectively. Information about antibodies is shown in Table 1.

### 4.2. Experimental Design and Drug Treatment

Information about the experimental design is shown in Table 2**:**

### 4.3. Behavioral Tests

Morris water maze test (MWM). The test was performed, as previously described, in a 120 cm round basin filled with water (24–25 °C; depth, 40 cm) with white paint added [12]. Briefly, there were two phases: In the training period, the mice were tested four times daily for 4 days with a hidden platform (8 cm diameter, 1.5 cm underwater). Probe trials, during which the platform was removed and mice were placed in the opposite quadrant, were assayed for 1 min. The data were obtained by a video tracking software called ANY-Maze (Stoelting Co., Wood Dale, IL, USA). All tests were performed from 8:00 to 12:00.

For the two-way shuttle (TWS) avoidance task, a shuttle box (30 cm × 30 cm × 20 cm) was divided into two equal sized compartments by an opaque partition with a channel. Both compartments were electrifiable and equipped with lighting equipment. Each mouse was put into the box with its back to the door and allowed to adapt for 5 min (walk freely, without light and foot shock), and then 30 cycle experiments were conducted. The experiment used light as the stimulation condition for 5 s. If the mice did not pass through the door to the adjacent chamber after an interval of 5 s, they were given a 0.3 MA, 5 s electric shock. The experimental interval was 30 s. The first day was the training period. Each mouse was repeated 30 times, and the TWS avoidance task was performed 72 h later. The test records include: the escape latency, the time when the mouse first moved to the adjacent chamber; shuttle times, including active avoidance times (shuttle to the other side after turning on the light) and passive avoidance times (shuttle to the other side after electric shock); and escape failure (the mice did not shuttle after being given a shock).

For the open field test, the mice were placed in a Plexiglas test device with a size of 50 × 50 × 40 cm. The device was divided into 16 equal squares, with the middle 4 squares forming the central area and the remaining 12 squares forming the peripheral areas. Each mouse was placed in the central area and recorded for 5 min. The recording software was ANY-Maze. The surfaces of the equipment were cleaned and dried after each tested mouse. The total distance traveled, the mean speed, and the time and distances in the central and corner were all recorded.

### 4.4. Mitochondrial Membrane Potential Assay

Beyotime Biotech provided the mitochondrial membrane potential (ΔΨm) assay kit (C2003S, Shanghai, China). ΔΨm was detected using a fluorescence microscope and flow cytometry. The cells were seeded in a 6-well plate. After the intervention, the collected cells were resuspended in 0.5 mL cell culture medium, added with 0.5 mL JC-1 working solution, gently mixed, and then incubated at 37 °C for 20 min. They were centrifuged 600× *g* at 4 °C for 3 min to precipitate cells. The cells were washed twice with 1 mL JC-1 buffer and analyzed using flow cytometry (Beckman Coulter, Brea, CA, USA). Or, the cells were washed once with PBS, and then 1 mL of medium and 1 mL of JC-1 working solution were added and incubated at 37 °C for 20 min. After that, the cells were washed twice with JC-1 buffer and observed with fluorescence microscope (RX50, SUNNY, Ningbo, China).

### 4.5. Reactive Oxygen Species (ROS) Detection

According to the instructions of the reactive oxygen species analysis kit (s0033s, Beyotime), dichlorofluorescin diacetate (DCFDA) was used as the probe to evaluate the intracellular ROS level. Briefly, incubation took place for 20 min at 37 °C with 10 μM DCFH-DA. Cells were washed three times with serum-free medium and then detected using a fluorescence microscope and flow cytometry (Beckman Coulter, USA).

### 4.6. Apoptosis Assays

*N2a* cells were seeded in a 6-well plate. After the intervention, 5 × 10^4^ cells were suspended in 195 μL binding buffer and incubated with 5 μL of Annexin V-FITC and 10 μL of PI at room temperature for 15 min in the dark. Flow cytometry (CytoFlex, Beckman Coulter, Brea, CA, USA) was immediately performed on stained cells.

### 4.7. Mitochondria (Mito) and Lysosome (Lyso) Tracker Staining

On a 24-well plate, *N2a* cells were seeded on cover slips. After treatment, the cells were incubated with 60 nM Mito-Tracker Green (C1048, Beyotime) and 50 nM Lyso-Tracker Red (C1046, Beyotime) for 30 min at 37 °C with 5% CO_2_. Lastly, the staining solution was removed; cells were added to fresh medium and observed by means of a fluorescence microscope.

### 4.8. Small Interfering RNA (siRNA) Interference

siRNAs were purchased from Ruibo Company (Guangzhou, China) and transfected according to the instructions. The PINK1 siRNA sequence was-CCAAGCGCGTGTCTGACCC-.

### 4.9. Transmission Electron Microscopy (TEM)

Hippocampal tissues and *N2a* cells were prefixed in 2.5% glutaraldehyde, and fixed in 1% osmium tetroxide with 1 mm^3^ size. Afterwards, the samples were dehydrated in acetone, infiltrated with Epox 812, and embedded. Sections were prepared and stained with citric acid lead and uranylacetate. Lastly, sections were examined under a JEM-1400-FLASH Transmission Electron Microscope (JEOL; Tokyo, Japan).

### 4.10. Mitochondrial Isolation and Western Blotting (WB)

Mitochondria were extracted with the mitochondria extraction kit of tissues (C3606) and cells (C3601) provided by Beyotime, and mitochondria and cytoplasmic proteins of Hippocampal tissues and *N2a* cells were retained for subsequent WB. Briefly, samples were lysed in RIPA buffer with PMSF and phosphatase inhibitor cocktail A (P1081, Beyotime) for 30 min at 4 °C, then centrifuged, and the supernatant was collected for SDS-PAGE. After blocking, the first antibody was incubated with the membranes at 4 °C overnight, followed by secondary antibody incubation for 1 h at 37 °C. Membranes were visualized on an image analysis system (Fusion, Germany). Antibodies are listed in Table 1**.**

### 4.11. Histological Analysis and Immunohistochemistry

The mice were anesthetized with 8% chloral hydrate and then perfused with physiological saline and 4% paraformaldehyde. Brain tissues were fixed in 4% paraformaldehyde for 3 days, embedded in wax and cut into 5 µm and 10 µm slices, which were xylene dewaxed, then ethanol dehydrated. Then, sections were stained for hematoxylin and eosin (H & E) and Nissl staining (C0117, Beyotime) according to the manufacturer’s instructions. Immunohistochemistry was performed as described [17]. The primary antibodies were incubated overnight at 4 °C, and secondary antibodies were incubated for 1 h at room temperature. A Leica DM500 light microscope (Leica Microsystems, Wetzlar Germany) was used to observe the histological sections.

### 4.12. Immunofluorescent Staining

Cells were fixed with 4% formaldehyde for 30 min and treated with QuickBlock™ Blocking Buffer (P0260, Beyotime) for 15 min at room temperature. Double-labeling immunofluorescent staining was performed to assess colocalization for PINK1 and Parkin, TOMM20 and LC3A/B. Incubation of primary antibodies at 4 °C overnight and of fluorescent secondary antibodies at 37 °C for one hour was performed. We purchased fluorescent secondary antibodies from SAB: anti-mouse IgG 488 (L3036), anti-rabbit IgG 594(L3017). Then, the fluorescent microscope was used to inspect the cells after the nuclear staining with DAPI (C1006, Beyotime). ImageJ/Fiji (National Institutes of Health, NIH) was used for image analysis in all the above experimental results.

### 4.13. Bisulfite Sequencing PCR (BSP)

The methylation of PINK1 promoter in hippocampus was detected by BSP. The MethPrimer (http://www.urogene.org/methprimer/, accessed on 11 June 2022) was used to design primers for PINK1 promoter, which was 5-ATTTTTAGGAGTTAGAGTTTYGGY-3 and 5- ACTCTACCTTTCCCRTAAACRTC-3.

### 4.14. Statistical Analysis

All statistical analyses were performed using the SPSS software version 21.0 (SPSS Inc., Chicago, IL, USA) and GraphPad Prism 8.3.0 (GraphPad Software Inc., San Diego, CA, USA), at least three times for each data set. Comparisons between different groups were made using one-way ANOVA with Tukey’s post hoc test. All data were represented as mean ± standard deviation (x ± s). *p* < 0.05 was considered statistically significant.

## Figures and Tables

**Figure 1 ijms-24-05896-f001:**
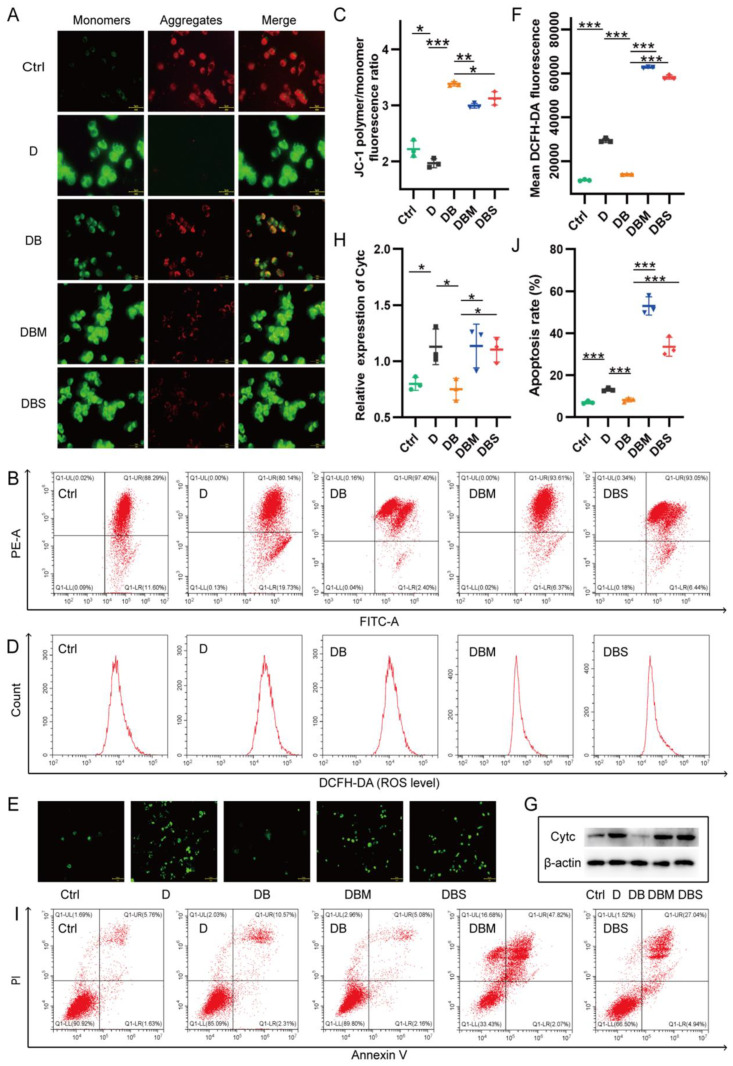
Berberine inhibits D-ribose-induced mitochondrial dysfunction by promoting mitophagy. Flow cytometry analysis of ΔΨm showed that the D group had weakened red fluorescence, enhanced green fluorescence, and a decreased ratio of the JC-1 aggregate/JC-1 monomer fluorescence compared with the control group (**A**–**C**). ΔΨm increased after the treatment with BBR (DB group), but decreased again after the addition of Midivi-1 (DBM group) or siRNA knockdown of PINK1 (DBS group). The results of ROS production (**D**–**F**), Cytc expression (**G**,**H**), and apoptosis (**I**,**J**) in each group showed the same trend as the results of ΔΨm. * *p* < 0.05, ** *p* < 0.01, *** *p* < 0.001. ΔΨm was determined using JC-1 staining with flow cytometry and IF (**A**–**C**). Scale bar = 5 μm. ROS measured by flow cytometry and IF (**D**–**F**). Scale bar = 10 μm. The cytochrome c in the cytoplasm of *N2a* cells was detected using WB (**G**,**H**). Detection of apoptosis by flow cytometry (**I**,**J**). Abbreviation: Ctrl, Control; D, D-ribose; DB, D-ribose + BBR; DBM, D-ribose + BBR + Mdivi-1; DBS, D-ribose + BBR + si-PINK1.

**Figure 2 ijms-24-05896-f002:**
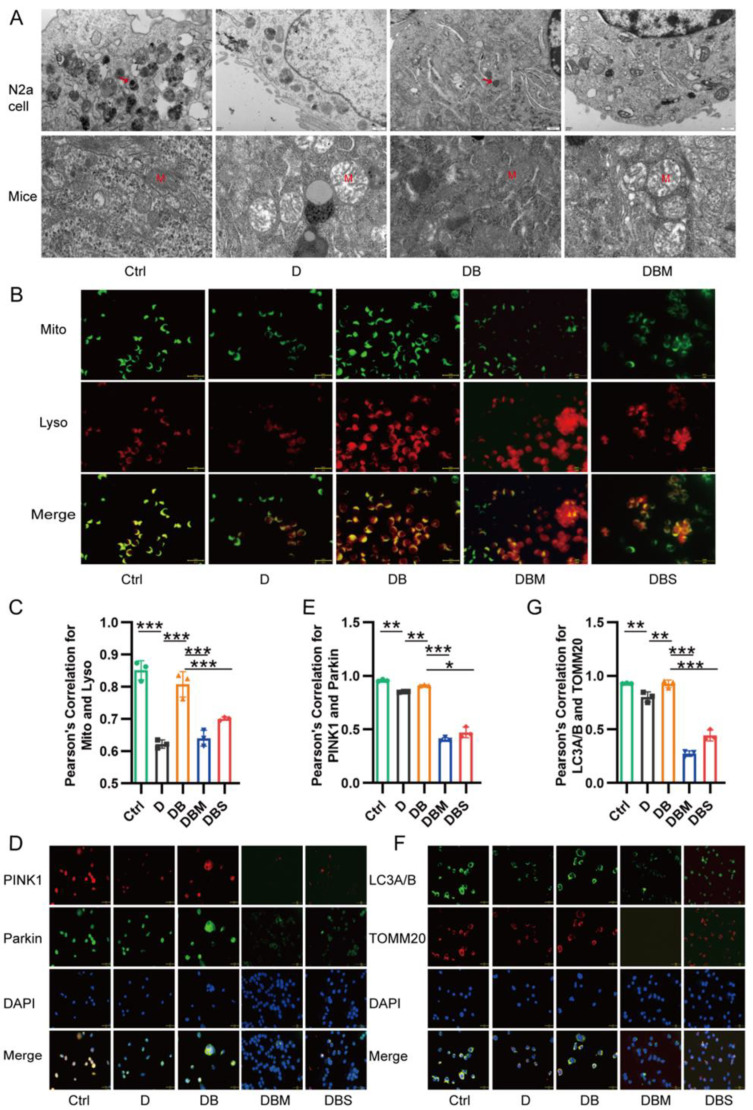
Berberine antagonized D-ribose-induced inhibition of mitophagy. Mitochondria of the D and DBM groups were swollen and cristae were ruptured (**A**). Berberine boosted the colocalization proportion of mitochondria and lysosomes (**B**,**C**). Compared with the Ctrl group, the colocalization of LC3A/B and TOMM20 (**F**,**G**), and PINK1 and Parkin (**D**,**E**) in the D group was significantly reduced, which blocked PINK1–Parkin-induced mitophagy. Electron microscopic evaluation of mitophagy in *N2a* cells and hippocampus. Red arrows indicate autophagic vesicles. M, mitochondria. Scale bar = 2 μm (**A**). *N2a* cells showing colocalization of mitochondria with lysosomes. Lyso: lysosome; and Mito: mitochondria (**B**,**C**). Immunofluorescence costaining of PINK1 and Parkin to detect colocalization (**D**,**E**). Immunofluorescence for LC3A/B and TOMM20 showing co-localization (**F**,**G**). Scale bar = 5 μm. * *p* < 0.05, ** *p* < 0.01, *** *p* < 0.001. Abbreviation: Ctrl, Control; D, D-ribose; DB, D-ribose + BBR; DBM, D-ribose + BBR + Mdivi-1; DBS, D-ribose + BBR + si-PINK1.

**Figure 3 ijms-24-05896-f003:**
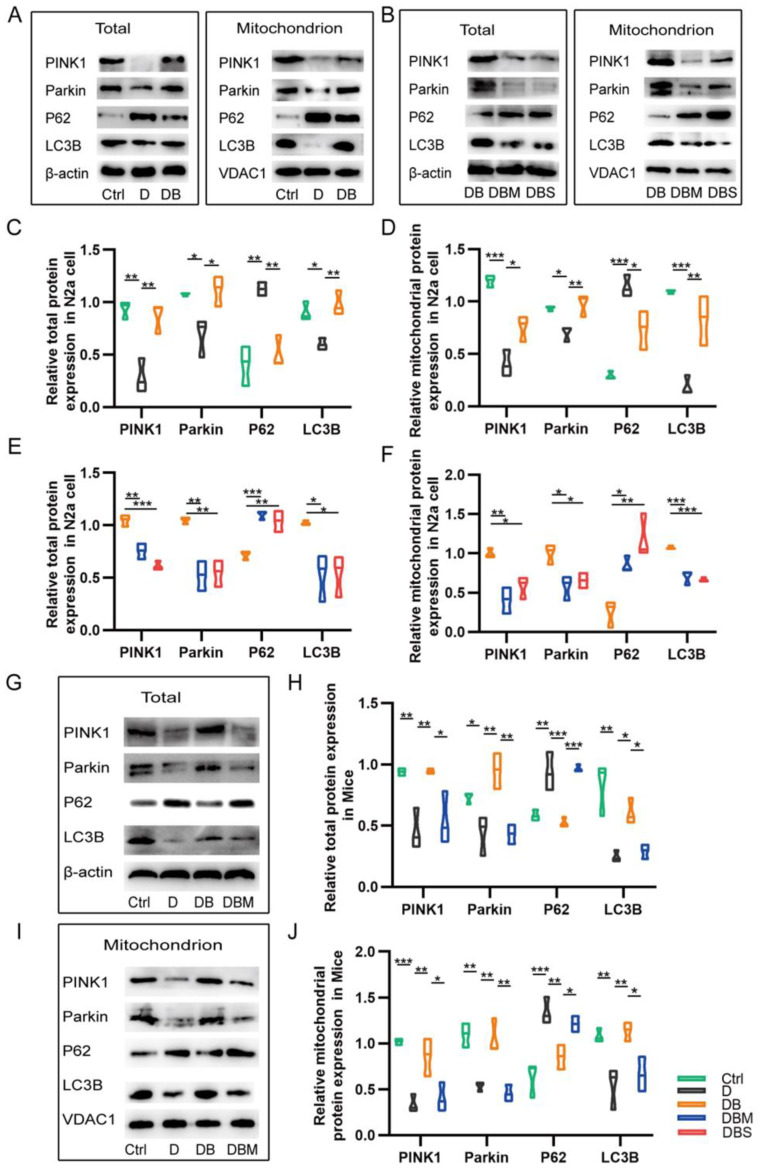
Berberine regulates the protein expression of PINK1–Parkin pathway proteins. D-ribose decreased the expression of key mitophagy-associated proteins PINK1, Parkin, and LC3B, both in vitro (**A**–**F**) and in vivo (**G**–**J**). Representative Western blot images and quantification of PINK1, Parkin, P62, and LC3B protein in *N2a* cells (**A**–**F**). Relative quantification of protein expression in *APP/PS1* mice by Western Blot (**G**–**J**). * *p* < 0.05, ** *p* < 0.01, *** *p* < 0.001. Abbreviation: Ctrl, Control; D, D-ribose; DB, D-ribose + BBR; DBM, D-ribose + BBR + Mdivi-1; DBS, D-ribose + BBR + si-PINK1.

**Figure 4 ijms-24-05896-f004:**
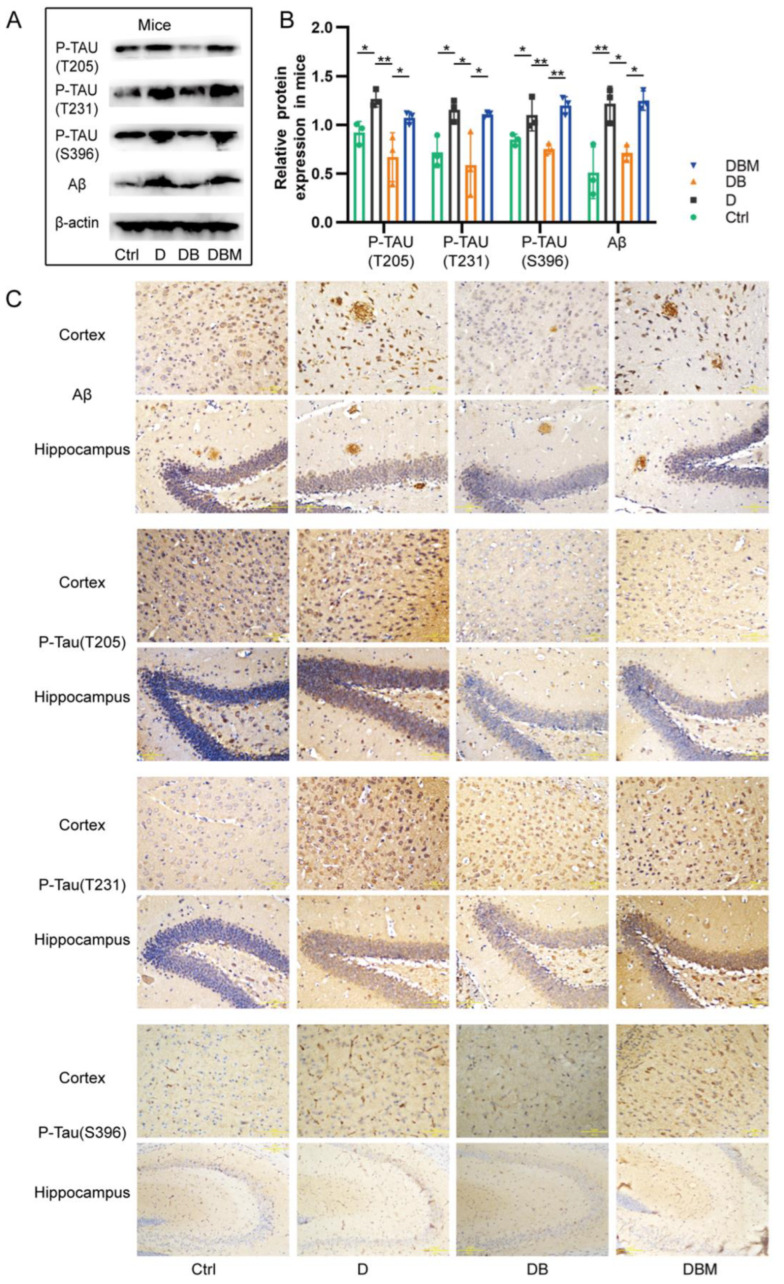
Berberine mitigated Alzheimer’s pathology induced by D-ribose by promoting mitophagy. The expression of Aβ, P-TAU (T205), P-TAU (T231) and P-TAU (S396) in the hippocampus and cortex of *APP/PS1* mice in the D group was significantly increased compared with the control group (**A**). The results of immunohistochemical staining also showed the same effect (**B**). Representative Western blot images and bar graph of p-Tau and Aβ in the cortex and hippocampus of *APP/PS1* mice (**A**,**B**). Representative images of immunohistochemistry of p-Tau and Aβ on sections of cortex and hippocampus (**C**). Aβ, p-Tau (T205), p-Tau (T231), scale bar = 5 μm. p-Tau (S396), scale bar = 10 μm. * *p* < 0.05, ** *p* < 0.01. Abbreviation: Aβ, β-amyloid; p-Tau (T205), phosphorylated T205 tau; p-Tau (T231), phosphorylated T231 tau; p-Tau (S396), phosphorylated S396 tau. Ctrl, Control; D, D-ribose; DB, D-ribose + BBR; DBM, D-ribose + BBR + Mdivi-1.

**Figure 5 ijms-24-05896-f005:**
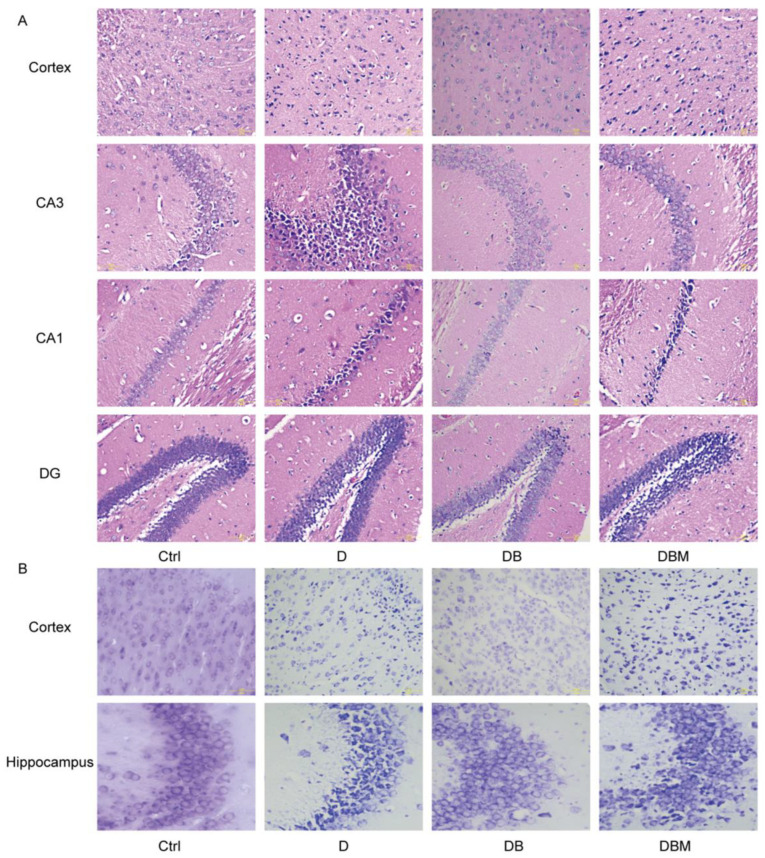
BBR rescued D-ribose-induced histological alterations. Histological sections of the cerebral cortex and the hippocampal stained with H & E and Nissl staining. Pictures showing the detail of H & E staining of the cerebral cortex and the hippocampal CA1, CA3 and DG neurons (**A**). Pictures showing the Nissl staining of the cerebral cortex and the hippocampal (**B**). Scale bar = 5  μm. Abbreviation: Ctrl, Control; D, D-ribose; DB, D-ribose + BBR; DBM, D-ribose + BBR + Mdivi-1.

**Figure 6 ijms-24-05896-f006:**
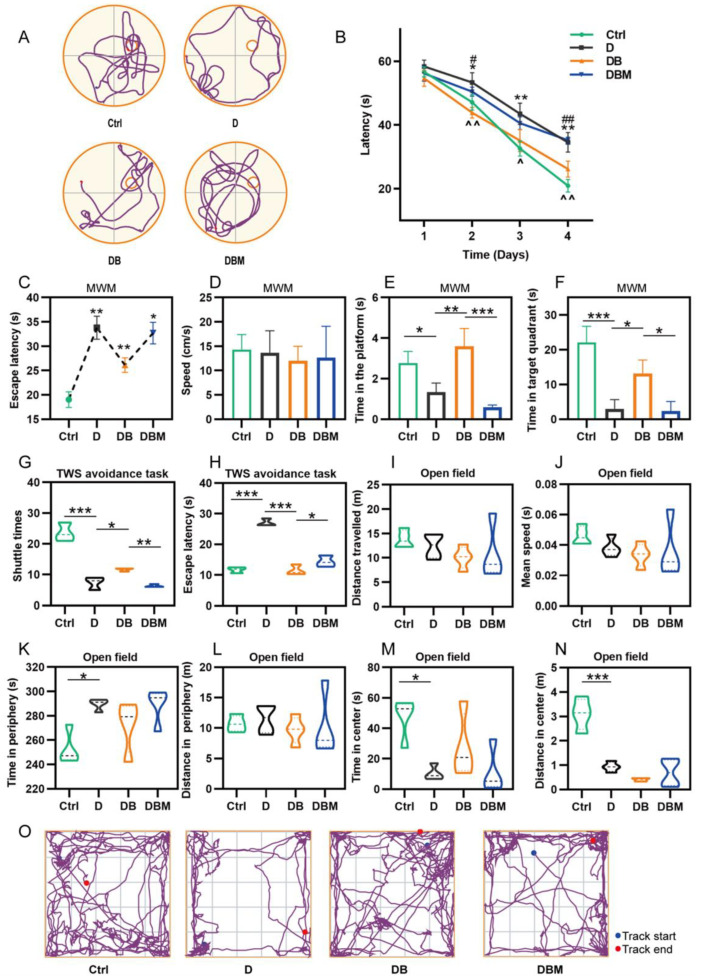
Effects of BBR and D-ribose on *APP/PS1* mouse spatial learning and memory and anxiety-like behavior. Compared with the control group, the D group had longer escape latency and shorter target quadrant time, which could be reversed by adding BBR; however, the DBM group showed similar results with the D group after adding Midivi-1 (**A**–**F**). Compared with the control group, the shuttle times in the D group were significantly reduced and the escape latency was increased. Berberine could significantly reverse the effect of D-ribose, but the DBM group showed fewer shuttle times and longer escape latency after the addition of Midivi-1 (**G**,**H**). Compared with the control group, the mice of the D group spent a longer time in the periphery and a shorter time and distance in center. The effect of D-ribose is not affected after taking BBR or Midivi-1 (**I**–**O**). Morris water maze (**A**–**F**). The representative swimming trace of the mice (**A**), avoid platform latency in visible (**B**) and hidden (**C**) tasks, swimming speed (**D**), time in the platform location (**E**) and time in the target quadrant (**F**). Two-way shuttle (TWS) avoidance task (**G**–**H**). Escape latency (**G**), the time when the mouse first moves to the adjacent chamber; Shuttle times (**H**), including active avoidance times and passive avoidance times. Open field (**I**–**O**). The total distances traveled (**I**) and the mean speed (**J**), the time and distances in the center and periphery (**K**–**N**) were recorded. The representative traces (**O**) were shown. *^,^#^,^^ *p* < 0.05, **^,^##^,^^^ *p* < 0.01, *** *p* < 0.001. Abbreviation: Ctrl, Control; D, D-ribose; DB, D-ribose + BBR; DBM, D-ribose + BBR + Mdivi-1.

**Figure 7 ijms-24-05896-f007:**
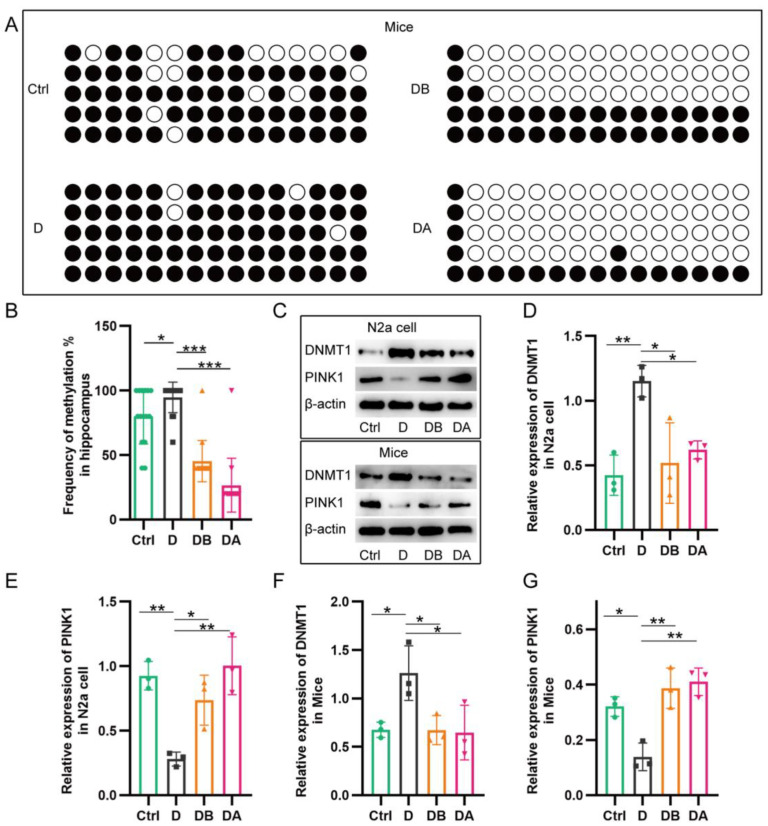
Effect of BBR and D-ribose on methylation of PINK1 promoter. Compared with the control group, the methylation of PINK1 promoter in the D group was significantly increased, which could be reversed by adding BBR, the same effect as DNA methylation inhibitor 5-Azacytidine (DA group) (**A**,**B**). In in vitro and in vivo experiments, compared with the control group, the D group had increased expression of DNMT1, but decreased PINK1. Compared with the D group, PINK1 was overexpressed; while DNMT1 was reduced in DB and DA group (**C**–**G**). BSP was used to detect the methylation level of PINK1 promoter in hippocampus of *APP/PS1* mice, dot plots and the statistical chart (**A**,**B**); Western blot was used to analyze the expression of PINK1 and DNMT1 in *N2a* cells and *APP/PS1* mice (**C**,**G**). * *p* < 0.05, ** *p* < 0.01, *** *p* < 0.001. Abbreviation: Ctrl, Control; D, D-ribose; DB, D-ribose + BBR; DA, D-ribose + 5-azacytidine.

**Figure 8 ijms-24-05896-f008:**
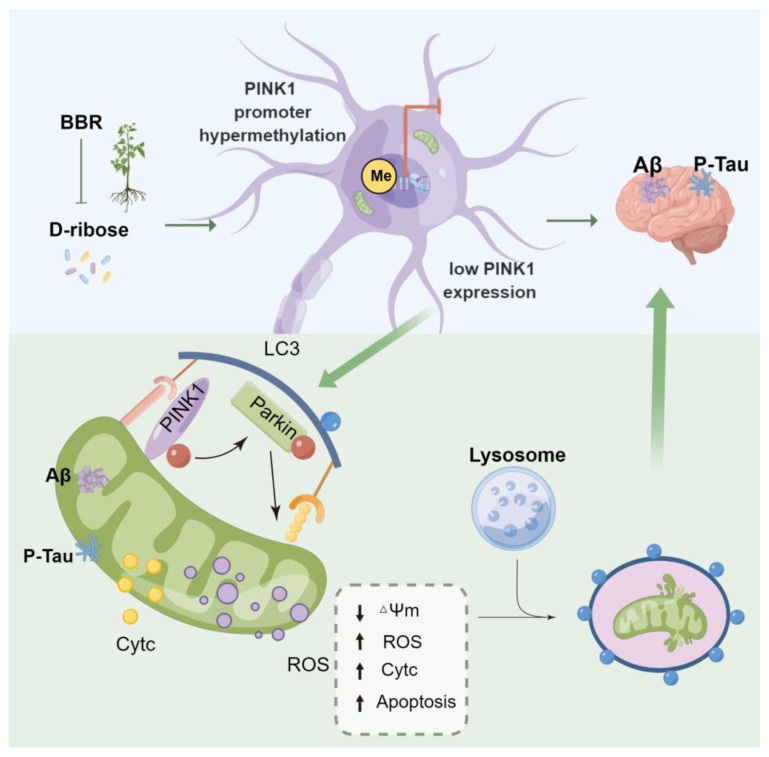
Hypothesis: proposed mechanism of BBR repairing D-ribose-induced Alzheimer‘s pathology. D-ribose induced mitochondrial dysfunction and damaged mitophagy including mitochondrial membrane potential decreased, ROS and Cytc release increased, cell apoptosis. However, BBR can reverse the above effects of D-ribose, improve mitochondrial function and restore mitophagy through the PINK1–Parkin pathway, so as to alleviate cognitive impairment and pathology induced by D-ribose, which is related to inhibition of PINK1 promoter methylation.

**Table 1 ijms-24-05896-t001:** The detailed antibody information.

Antibody	Species Source	Supplier	Identifier	Dilution		
				WB	IHC	IF
SQSTM1/p62	Rabbit	CST, Danvers, MA, USA	39749	1:1000		
PINK1	Rabbit	Proteintech, Chicago, IL, USA	23274-1-AP	1:1000		
LC3A/B	Rabbit	CST, Danvers, MA, USA	12741	1:1000		1:500
LC3B	Rabbit	CST, Beverly, MA, USA	3868	1:1000		
β-actin	Mouse	SAB, College Park, MD, USA	48139	1:2000		
TOMM20	Mouse	Abcam, Cambridge, MA, USA	ab283317			1:500
PARKIN	Rabbit	Proteintech, Hubei, China	14060-1-AP	1:1000		1:100
VDAC1	Rabbit	SAB, College Park, MD, USA	52646	1:1000		
Phospho-Tau (Thr205)	Rabbit	CST, Danvers, MA, USA	49561	1:1000	1:600	
Tau (phospho T231)	Rabbit	Abcam, Cambridge, MA, USA	ab151559	1:5000	1:500	
Tau (phospho S396)	Rabbit	Abcam, Cambridge, MA, USA	ab109390	1:10,000	1:1000	
β-Amyloid (1-42)	Rabbit	CST, Danvers, MA, USA	14974	1:1000		
beta Amyloid	Rabbit	Abcam, Cambridge, MA, USA	ab201060		1:1000	
PINK1	Rabbit	SAB, College Park, MD, USA	29297	1:1000		
PARKIN	Mouse	CST, Danvers, MA, USA	4211	1:1000		
DNMT1	Rabbit	Beyotime, Beijing, China	AF5150	1:500		
Cytochrome	Mouse	Beyotime, Beijing, China	AC909	1:200		
PINK1	Mouse	Bioss, Beijing, China	Bsm-51265M			1:100

**Table 2 ijms-24-05896-t002:** Treatment groups and doses.

Groups	*APP/PS1* Mice	*N2a* Cell
Control (Ctrl)	PBS, intraperitoneally	No treatment
D-ribose (D)	D-ribose, 2 g/kg/d, 30 days, intraperitoneally	D-ribose, 10 mM, 24 h
D-ribose + BBR (DB)	D-ribose (same as above); BBR, 100 mg/kg/d, 30 days, oral gavage	D-ribose (same as above); BBR, 1 μM, 24 h
D-ribose + 5-azacytidine (DA)	D-ribose (same as above); 5-azacytidine, 10 mg/kg/d, 3 times/week, final 3 weeks, intraperitoneally	D-ribose (same as above); 5-azacytidine, 10 μM, 24 h
D-ribose + BBR + Mdivi-1 (DBM)	D-ribose (same as above); BBR (same as above); Mdivi-1, 1 mg/kg, every other day, 30 days, intraperitoneally	D-ribose (same as above); BBR (same as above); Mdivi-1, 25 μM, 24 h
D-ribose + BBR + si-PINK1 (DBS)	-	D-ribose (same as above); BBR (same as above); si-PINK1, 50 nM, 48 h

## Data Availability

Data is contained within the article.

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
