# Peer review of "Berberine Rescues D-Ribose-Induced Alzheimer‘s Pathology via Promoting Mitophagy"

_ijms, 2023, doi:10.3390/ijms24065896_

Round 1

Reviewer 1 Report

The authors of this manuscript studied the effects of BBR on mitochondria function influence by  D-ribose in the  AD mice model and  cell-cultures. The topic and obtained findings are exciting. Nevertheless, I would suggest the following recommendations.

The method, results and conclusion parts of the abstract should be better elaborated.

Formulation of study aim at the end of the Introduction is not precise, state clear aim what was planned to achieve using that methods and tools.

It is not clear in the  Method part ‘Research on animals has been approved by the Univer-350 sity of Chongqing Medical’s Committee of Ethics’ stated wheather animal research in general or for this certain study is approved. Which international guidelines for 3Rs and animal care are followed in this study?

Method section is missing in which solution D-ribose(D) and B are diluted.

Why D-ribose(D) goup mice received D-ribose(D) intraperitoneally for 30 days? But D-ribose+BBR (DB) was given in mice by oral gavage for 30 days? Add a purpose for this difference in the text. It is not stated how for the control group mice Control(Ctrl) is PBS given (intraperitoneally or by oral gavage)?

The first paragraph of Discussion is missing references.

Minor: The whole text needs to be revised, the spaces in the text between words and numbers, between sentences must be complied.

What is reference for the Morris water maze test? Mentioned is ‘The test was performed according to previously described..’  no reference added. How long is training session for mice?

Reviewer 2 Report

The current paper is additional research on berberine. The current paper examines it as a treatment for AD. The paper needs some appropriate English editing. Several sentences are run on sentences. In appropriate punctuation. Several of the figure captions need to be expanded. 

Reviewer 3 Report

The manuscript needs extensive revision based on the following comments. 

1. The present form of each figure is not easy to understand. Thus, the Authors should be revised all the statistics into bar graphs. 

2. Figure 4 is not discussed at all. So the results and discussion part of figure 4 should be included. 

3. Figure 3A and B. I am wondering why the authors split the WB data. The protein expression of the DB group in the total and mitochondrial fractions is different in both figures. Please address why it is different. Also, the original blot is very difficult to understand. Please correct all the blot images.   

4. Figure 8 does not represent any sense. So redraw the figure to point out the possible mechanism of action of Berberin that rescues D-ribose-induced Alzheimer's pathology through activating mitophagy.    

Round 2

Reviewer 1 Report

English style editing is needed.

Reviewer 3 Report

The authors provided all my queries satisfactorily.